# Sowing Seeds to Harvest Healthier Adults: The Working Principles and Impact of Participatory Health Research with Children in a Primary School Context

**DOI:** 10.3390/ijerph17020451

**Published:** 2020-01-10

**Authors:** Tineke Abma, Sarah Lips, Janine Schrijver

**Affiliations:** 1Department of Medical Humanities, University Medical Centre, 1081 HV Amsterdam, The Netherlands; s.lips@amsterdamumc.nl; 2Art Collective B.A.D, 3082 MG Rotterdam, The Netherlands; janineschrijver@xs4all.nl

**Keywords:** children, creativity, empowerment, impact, participatory action research, primary schools

## Abstract

Participatory research on health-related topics with children is promising but current literature offers limited guidance on how to involve children and falls short on the reporting impact. The purpose of this article is to heighten our understanding of the working principles and impact of participatory health research (PHR) with children. We completed a PHR project in two primary schools, which included children from a multiethnic, deprived neighborhood in the second largest city in The Netherlands over a period of three school years (2016–2019). The impact on the children’s subjective health has been measured via process evaluation using qualitative and quantitative methods from the perspectives of all involved (children, their teachers, parents, and community partners). The main working principles included: Experiential learning; addressing uncomfortable issues; stepping outside your environment; and keeping it simple. Participatory actions valued most by the children included: Walking tours, photovoice, foodlabs, sportlabs, and to a lesser extent: Making a newspaper, mindfulness, and Capoeira. The project reached and engaged many children, parents, teachers, and community partners into healthy lifestyles and broadened and deepened the children’s awareness and understanding of health behavior. ‘Sowing seeds’ is the metaphor that captures the broader impact of this project: Planting seeds to harvest healthier adults.

## 1. Introduction

Participation and participatory action research are increasingly recognized within the field of health promotion as pathways to the recognition of potentials and resources available to citizens and communities to improve their health and well-being [1,2,3,4,5]. One of the principles underlying participatory action research is to maximize the participation of people whose life or work is at stake [5,6]. Participation is intrinsically valued to counter epistemic injustice [7], and considered a way to gain a better understanding of people’s lifeworld [8]. While striving for the equal co-creation of knowledge, participation may range from a one-way consultation to involvement in all stages of the research through processes of democratic decision-making and shared control [6].

Participatory action research has been practiced with various, often marginalized groups, such as older people [9,10] and people who have psychiatric vulnerabilities [11]. Moreover, children have been engaged in participatory research to gain a better insight into their lifeworld and develop health promotion action plans that better meet their needs [12,13,14]. However, in some instances participatory studies aiming to improve the health and well-being of children are actually carried out by adult representatives of children or advocates instead of children themselves [15]. Such studies may underestimate the capacity of children and turn out to be less relevant. They also raise the question of what kind of methodology is needed to engage children in a way that appeals to them, and recognizes power asymmetries between adults and children. 

It is expected that participatory research with children can impact children’s lives at personal, familial, communal, and institutional levels and can even influence global issues [16,17,18]. Yet, the actual impact of participatory action research with children has not yet been well addressed in the literature. Moreover, there is still a lack of good insight of the impact of participatory action research in the fields of health and health promotion in general. Cook et al. [19] completed a systematic review of the impact of participatory health research and concluded that though not much has been published in this area, what exists has made a difference with, within, and for those who are engaging in the research. It may well be that many projects are small and local, and do not find their way into academic publications. Moreover, the impact may be difficult to measure objectively. Furthermore, impact may include both tangible (for example, weight loss) and intangible outcomes, such as heightened self-esteem, awareness, perceived health, and well-being. 

Our conclusion is that the literature still offers limited guidance on involving children in research to address child and youth health issues [15], and falls short on reporting impact (tangible and intangible) of participatory research on children’s lives. This paper therefore aims to contribute to the understanding of working principles and impact of participatory health research with children. Working principles concern meaningful ways to involve children in health-related research. Impact refers to the changes in the lives of the children and their local communities. Abma et al. describe impact in the context of participatory research: ‘In participatory research change is usually recognized as the contribution that research makes to the people involved in the research and the communities and organizations they are part of, although change on the wider society, policy making and the academy may also occur’ [6] p. 101.

We begin with the study methodology, proceeding to the results to enrich our understanding of the working principles of PHR with children and impact on their health and well-being. In the discussion, we reflect on the various types of impact that PHR can bring about and the working principles that can strengthen impact. 

## 2. Materials and Methods

In this section, we describe our approach and theory on participatory health research (Section 2.1), the context and emergence of the KLIK program (Section 2.2), and the materials and methods (Section 2.3). KLIK is a Dutch acronym for Kind Leert Inventief Kracht, it stands for Children Learn Inventive Strength.

### 2.1. Approach and Theory on Participatory Health Research

Our participatory research approach is grounded in the critical pedagogy of Paulo Freire [20] and related approaches, including Action Research [21], Participatory Action Research [22], and Community-Based Health Research [2,23]. All of these approaches aim for social transformation through action-reflection-learning cycles within communities. The purpose is to heighten people’s understanding of their situation and gain control over it, and as a result become more capable of taking actions and developing the necessary skills (e.g., interpersonal and negotiation) to collectively improve their situations. Guiding principles include mutual respect, democratic decision-making, maximum participation and social inclusion, mutual learning, making a difference, and collective action [5,24]. Maximum participation guides the methodological choices in participatory research, and often includes creative and arts-based methods to involve those who are less verbally oriented [6,25]. 

Our research approach also resembles critical teaching in schools, as well as critical pedagogical approaches to health and physical education [26,27]. Critical pedagogy departs from the notion that the teacher has a crucial role in either reproducing the status quo in society or facilitating critical consciousness to stimulate children to question their world, to raise critical issues, to take control over their lives, to resist power and dominant constructions of reality, and to destabilize stereotypes. Critical pedagogy acknowledges the complexity of children’s lives and structural inequalities [27]. It is critical of attempts to make children individually responsible for their health and well-being because this ignores the structural inequalities that directly impact their quality of life and health. It is concerned by the strong push for health and physical education that has resulted from the so-called ‘obesity epidemic’ as this might medicalize and exclude certain children and undermine their human-ness [28].

### 2.2. The Context and Emergence of the KLIK Program

KLIK was the name of our participatory health research project with children in two primary schools. KLIK stood for the click of the camera, because the project involved giving the children a camera to explore their environment, and the click with yourself and your surroundings. KLIK aimed to work with (versus on) children on health-related issues. KLIK was set up by two of us (first and last author) in the suburb of Rotterdam where we both have lived for more than thirty years. We are both mothers with grown-up children, and feel quite privileged with good jobs and healthy families. Rotterdam is the poorest city in The Netherlands, and one out of four children grow up in poverty [29]. Our goal was to contribute to the well-being of young children who are less well off by combining our capacities and networks in the science and art fields. One of us, the first author, works at a university as a professor in the field of participatory research. The last author has a background in the arts and works as a professional photographer. Together we formed a core research and intervention team. We were assisted by a junior researcher from the university. The core team remained stable over the years, and this contributed to the trust developed with the children and community partners.

KLIK was set up in an underprivileged neighborhood: Oud-Charlois, on the South-bank of Rotterdam, one of the largest cities in The Netherlands. Its population was relatively young, culturally mixed (Moroccan, Turkish, Surinamese, and lately also East-European), low-educated, with high rates of unemployment and poverty. In the area where KLIK was based, many children did not have access to a sporting club or the resources to pay for a membership. Many parents were facing difficulties to make ends meet. The neighborhood could be characterized as ‘obesogenic’ with an excess of cheap, unhealthy food and an environment that did not stimulate children to exercise [29]. The population was relatively unhealthy with higher prevalence of obesity and chronic diseases [30,31]. Public schools had difficulty finding qualified staff, and many teachers felt overburdened by large caseloads of children in need of extra help and support given their home situations (financial debt, emotional neglect, etc.). At the same time, a lot of artists and creative entrepreneurs were attracted to this area because of cheap rental properties.

At the beginning of the project, interviews and meetings with all community partners (schools, healthcare and welfare providers, dieticians, sport and cultural coaches, migrant self-organizations, scholarly artists) were held, to develop a mutual relationship and dialogue, to discover and formulate basic principles for the approach that would be fundamental to all activities developed within the project, and to set priorities. The topics covered were experiences of the neighborhood, the health of residents, in particular children, experience of health promotion schemes, ideas for health promotion, potential partnerships, and possible obstacles to or opportunities for health promotion in the neighborhood. Moreover, we started to experiment in one primary school with (participatory research) methods appropriate for children. One of these methods was photo-voice [32], a technique in which children received cameras to explore their life-worlds and take pictures of their surroundings. Photovoice was combined with photography by the researcher to visualize the broader life-world context of the children, including their neighborhood, a method which resembles visual anthropology [33]. All these activities and gatherings with stakeholders led to a formulation of a research proposal (2014–2015). After funding approval, four classes participated over the course of three school years, from 2016 until 2019. The classes had approximately 20 children each and the children were in grade four (8–9 years old) at the start and grade six (11–12 years old) in the third year. In total we reached 80 children. We started KLIK with a global idea based on the health promotion literature that living a healthy life is related to healthy nutrition and exercise, but also to mental well-being. For many families in the neighborhood in which KLIK was developed, attaining health in all of these (related) aspects was especially challenging. Departing from this global idea, we searched to develop a program without imposing our own ideas and preferences, originating from our perspective as privileged (well educated, healthy, and wealthy) citizens. We therefore sought to engage those whose lives and work were concerned, enabling us to attune to their needs and preferences. The specific actions in KLIK were thus developed throughout the project and in an ongoing dialogue with the teachers, parents, community partners, and the children themselves. 

In doing so, our focus in the first year was on healthy food. This was partly guided by concerns of teachers, who observed that the children arrived at school without having had breakfast, thus becoming hungry during the lessons. They also saw them bringing unhealthy lunch boxes or buying the same snack food every day. One of the teachers noticed that children complained about abdominal pain and had difficulty going to the toilet. Another teacher pointed out that the treats children were bringing to celebrate their birthdays were generally unhealthy. In response, a course was developed in which the children could experiment with making healthy, yet festive and enjoyable treats. In the second year, we concentrated on exercise and sport because we noticed that the children associated being healthy with fruit and vegetables but did not link it to sport and exercise. We also offered them a training mindfulness in year two because we observed that many children were having trouble maintaining their attention and concentration in class. In year three, we aimed for the repetition and integration of all knowledge (developed during years one and two?). See Appendix A for an overview of all activities carried out over the three years. See Appendix A: Activities with the children in the KLIK program 2016–2019 (Appendix A).

### 2.3. Materials and Methods

During KLIK activities, children were asked to express themselves via arts-based methods like photovoice, game-playing, mind-maps, and drawings. These symbolic tools which included other means of expression in addition to language enabled the children to express themselves and generated rich insight’s in their perspectives. The photos and other creative artefacts were used to elicit meaning [34,35,36]. Participant observation took place during all activities and an extensive observational diary was kept. This included information about social (inter) action, including group dynamics during project activities in class, interactions between school children, and teachers. It also provided an opportunity for informal conversation. One researcher, being a professional documentary photographer, captured the process in photos.

Data were analyzed interpretatively and crosschecked by the researchers [32]. The analysis had a cyclic and iterative character. We started by reading and rereading the extensive amount and multifaceted material (transcripts, field notes, logbooks), as well as reviewing and interpreting the generated creative artifacts (drawings, mind-maps, photos). This process included many conversations within the research team, during which time we discussed our initial ideas about the impact and working principles of KLIK. We used a hermeneutic approach and focused on the meaning of experiences and social action. To enrich our sense making and validate our findings, we discussed our findings with various partners, including the children and teachers in school, which also offered them the opportunity to reflect and give feedback (participatory member check). Independent evaluators and peers helped us in this process by asking evaluative questions about KLIK during the analysis stage. A quantitative evaluation was completed by an external research organization specialized in exercise and sports. They developed a short online survey for the children and their parents to measure their perceived fitness and health. A child-friendly method was used to engage children: They were invited to respond to simple questions using smileys. The most important dependent variable ‘experienced health’ was measured by a visual scale with five smileys from sad to cheerful. Other questions focused on experienced fitness, perceived body image, and food and sport/exercise behaviour. In the premeasured phase, children and parents were also asked what they wanted to learn about healthy lifestyles, to describe what they think works best, and what they need to live healthier lifestyles via open questions. In the post measurement phase, the children and parents responded to evaluative questions about KLIK that sought to find out how they experienced the activities and what they learned.

The project was financially sponsored by FondsNutsOhra (FNO), a charity fund in The Netherlands, as part of their program A Healthy Future Nearby. According to the Medical Ethics Review Committee of VU University Medical Center (registered with the US Office for Human Research Protections as IRB00002991; FWA number: FWA00017598) the Medical Research Involving Human Subjects Act (Wet op Medisch Onderzoek, WMO) did not apply to KLIK: Niet WMO-advies METc verklaring VUmc 2016.582. In addition to the informed consent and confidentiality, various additional ethical principles were taken into consideration during this project: Working on mutual respect, participation, active learning, making a positive change, contributing to collective action, and personal integrity [24]. Approval was obtained for the activities and the publication of all photos used for publications. This included a process of negotiation with the children and their parents. We respected the children and parents’ will not to join activities and release certain images. Ethical guidelines as well as dedicated time within our research team meetings and conversations with critical friends and peers were helpful to discuss issues of power, ethics, and responsibilities [32].

## 3. Results

We initiated KLIK as an alternative to current plans and policies to tackle the ‘obesity epidemic’ [28,37]. Health promotion projects are often driven by adults determining what children need and as a result are dislocated from the lived experience of the children themselves [38]. If children do not know why they need, for example, to do exercises or eat more healthily, they will never be intrinsically motivated to change their own actions and behavior [39]. Moreover, information and knowledge is not enough to change behavior [39]. Integrating knowledge into one’s life requires new habits and routines, and an enabling environment that stimulates healthy behavior. Below we present the results of our process evaluation in the following order: Impact (Section 3.1) and working principles (Section 3.2).

### 3.1. Impact

In participatory research, impact can be defined in terms of the changes that take place in the lives of those whose life or work is the subject of the study. In our case, the school children’s health and well-being. Impact can include concrete, visible phenomena such as losing weight, new habits, and behaviour, as well as intangible changes, such as improved awareness and understanding or heightened self-esteem and pride. The results from our study demonstrate increased knowledge, awareness, and repertoire of skills in relation to health and well-being. The KLIK lessons not only imparted knowledge, but also taught the children new ways to take care of themselves, to explore and inquire, and to experience what it means to live healthy and well, which they would otherwise not have encountered in their lives. In the program evaluation, approximately 53% of the children reported they had learned new insights through KLIK.

What the kids appreciated most about KLIK was its ‘special’ nature; the approach to teaching was quite different from the regular activities and lessons in school. The quantitative evaluation showed their appreciation of the activities. They highly ranked the visit to the Corpus museum, the food lab, and the photography (see Appendix A). These quantitative figures corroborate the results of our qualitative evaluations of the process and activities over the years.

Below are field-note extractions of our evaluative conversations with children, teachers, parents, and community partners, which illustrate what the KLIK activities brought about in terms of reaching and engaging many children, parents, teachers, and community partners into healthy lifestyles (Section 3.1.1) and broadening and deepening the children’s awareness and understanding of health (see Figure 1; Figure 2 and Figure 3).

#### 3.1.1. Reaching and Engaging many Children, Parents, and Community Partners into Healthy Lifestyles

Health promotion is often associated with normative guidelines passed on from experts to lay people who need to adjust their way of living to become and stay healthy. This can generate negative emotions because it includes prescriptions (do’s and don’ts). In retrospect, we can see how KLIK activities made health and well-being an interesting topic associated with positive emotions for children, parents, teachers, and other community partners.

At 2 p.m. we enter Group 6. The children are happy to see us. Two boys in the front joke that we smell like vegetables. We are welcomed by an extended ‘Hello misses KLIK’. When we announce we are going to teach them for three years, all of them yell and shout. They even get louder when we say that we are going outside to take pictures. One of the boys says, ‘Finally something fun’. The children receive cameras and are instructed to take photos of their neighborhood using a little mirror. When we come back in class many of them report they really liked using the mirror. One of the girls points to a smiley. All enjoy sharing and talking about what they have seen: Trees, people, cars, trash, shit, etc.(Logbook, 31 January 2016)

We talk with two teachers about the past year, and one of them says: ‘The children really liked it’. The other says some of the children in class do not like anything, but that one of those guys was actively engaged in making a vegetable and fruit mask today. Additionally: ‘They are really disappointed when it is postponed now and then’. The teachers share what works well: Being active in class because quite a few of the kids have trouble sitting and listening all the time. They notice that the children are more aware, also of their own talents. They mention that the children take what they learned back to their families and put it into practice, for example the ‘schijf van vijf’ (official food guide communicated by the Dutch nutrition center; infographic in the form of a sort of pie chart), and the foodlab where they tasted colored vegetables and all ran around with blue tongues. The teachers sensed that initially some parents were a bit skeptical, but those are now the ones who say they hear nice stories about KLIK.(Logbook, 30 May 2017)

In KLIK we collaborated with the teachers of the two participating schools, and also engaged many other neighborhood community partners, in order to extend our social base and benefit from their expertise and networks. Many of them were committed and willing to help us reach children and their families, also outside of school and during holidays. One of those organizations was TOS (‘Thuis Op Straat’) an organization of coaches on the street. TOS focused on playing games and social activities such as soccer and dance competitions. TOS coaches were interested in integrating ideas on health and well-being because they noticed that more and more kids and teenagers were staying inside their homes gaming (‘When I was young we played outside, now we are happy if there are four kids on the square’.), consuming drinking energy drinks, smoking (they estimate 60% starts smoking at the age of 13), and buying snack food (16 January 2017).

During a “Beach Week,” which TOS organized during the summer break for kids in the neighborhood who could not go on vacation, we jointly developed a healthy-treat workshop. The collaboration with TOS had started earlier with the street coaches (called ‘big brothers’ and ‘big sisters’) taking part in role-playing exercises to see how they could become role models for the younger kids to draw their attention to healthy food and drinks and consider possible dilemmas, such as talking with their parents about unhealthy lifestyles. During the Beach Week, TOS organized a workshop on making healthy treats with fruits and vegetables with our help and that of a local dietician. Afterwards, the dietician wrote us an email describing what struck her:
Some children looked starved and lacking nutrition, which she recognized in her own practice: ‘I see kids in my practice with hollow eyes’.Everything, really everything, that was on the table was picked up by the kids and their parents. A parent asked: ‘Can I have a piece of cucumber for my kids?’ (the cucumber was a decoration in the form of a crocodile) (see Figure 4).A bunch of very creative kids came to the workshop.A group of enthusiastic TOS colleagues made it fun for the kids.(Logbook, 17 July 2017)

Through KLIK, we deliberately sought to develop a large variety of activities and to conduct these in different environments, in and outside the schools. One reason why this was done was to enable the children to tap into, or even newly discover, a broad range of personal skills, knowledge, and preferences. Additionally, this was also important in order to reach (metaphorically speaking: To plant a seed in) as many children as possible. We soon noticed that every activity did not appeal to every child all of the time, but because of the large variation, each and every one came across something they liked at some point. Mindfulness was, for example, made many of the children uncomfortable, yet several of the kids appreciated it as one of the highlights of KLIK (‘I felt relaxed and want to do it more often. That lesson was meant to create a KLIK with yourself’.). Moreover, the children were enriched with a lot of activities that they would not have encountered otherwise. At the end of the three school years, a series of group interviews with the children and their drawings of what they had learned revealed their appreciation of KLIK (see Figure 5):
‘Nice memories’.‘I was often critical, so it may come as a surprise, but actually I liked KLIK very much’.‘I learned about my body and how I do things’.‘Own initiative’.‘That you just need to try out things’.

#### 3.1.2. Deepening and Broadening Their Awareness and Understanding of Health

Over the three years, the kids deepened their awareness and understanding of what it means to live healthy and well. They were increasingly able to interpret situations, to see patterns (in behavior?), and to contextualize the knowledge newly learned in KLIK by linking it with situations happening outside school. The following example shows how the children related their knowledge to a situation wherein one of their teachers died of cancer:
On our way home at the bus stop I met and talked with one of the teachers. She tells me that one of the school teachers has died, and that everyone is focused on this tragic event. It was known for quite some time that she was terminally ill. What strikes the teacher is that the kids are really supporting each other. She also tells me how she referred to our visit to Corpus (a museum about the body) to explain the deceased teacher’s illness and how the kids were able to link this experience to what they learned in KLIK. (Logbook, 29 January 2019)
One of the other teachers mentions how the Corpus visit helped her to talk with the kids about the death of her colleague. She smoked and the kids now understand what this can do to one’s health. The teacher also tells us how some of the children were angry: ‘Why did she smoke?!’ (Logbook, 5 February 2019)

As the teachers mentioned, doing new activities also revealed hidden talents (such as having an eye for composition, agility, a green thumb), and improved the children’s self-esteem. It also inspired the teachers to try out new things with the children whom they sometimes underestimated. For instance, one of the teachers started asking the children to write little creative stories that invited them to share their experiences with KLIK for a newspaper made by and for kids about health and well-being. The teacher also noticed how the students spontaneously started interviewing each other, a skill they had practiced in the KLIK lessons for developing a newspaper. Parents appreciated the activities because many of them could not offer such opportunities to their kids.
‘A bit more aware, looking after calories in soda drinks’.‘Yeah, I am attending more to my behavior’.‘Yes, because I do not want to grow fat, so you have to watch what you eat’.

Others said they were already knowledgeable—knowing what is healthy and unhealthy—but that they had gained more practical insights on how to take care of their bodies and live healthily. In response to being asked what they would carry along later in life, they replied:
‘Being relaxed’‘Everything!’‘Keep on exercising’.‘Caring for my body’.‘Healthy fruit’.‘Healthy food in the canteen’.‘Enough sleep’.

Looking back on all the lessons learned during the program, the children could remember them and explain why these were important for staying healthy and well:
‘Yes, to know how the digestion of food works, how you can keep yourself relaxed, and what is healthy and unhealthy, and what belongs to the ‘schijf van vijf’ (Food Guide)’.

Therefore, while measurable changes were hard to objectify, we saw that seeds had been planted. The fact that the KLIK lessons differed from the regular classes, which was explicitly mentioned by the children, makes it more likely that they will remember the lessons learned:
‘If I would have to explain to someone what KLIK is about, I would tell that it is primarily about your health and that you do experiments on food and exercise. (…) I would like other children to get KLIK lessons, because they could also experience the fun and learn new things. I think that I especially take with me the pleasant memories’.(Boy, May 2019, KLIK Newspaper)

### 3.2. Working Principles

Based on the analysis of all data (qualitative and quantitative), we distilled four working principles to engage and motivate children to participate in action research for health and social well-being. These are presented below: (1) Experiential learning; (2) addressing uncomfortable issues; (3) stepping outside your environment; and (4) keeping it simple.

#### 3.2.1. Working Principle 1: Experiential Learning

We invited the children to explore, experiment, and literally taste and feel through ‘labs’ and experimentation, and gave them the confidence to be able to do this. From the beginning, we focused on experiential learning and tried to build more horizontal relationships in which the adults did not stand above the children but respected their knowledge and perspectives. The project engaged children in playful and creative activities to broaden their horizons so that they could explore what health and healthy living meant for them. Through this approach, we did not start from the normative position of telling them what they should do to stay or become healthy. Instead, we engaged them in inquiry and activities through which they could experience their embodied beings. The children themselves became researchers of their bodies and lives. Symbolically, we emphasized this by giving them white lab coats to wear during activities. The lab coat symbolized the researcher/expert.

The choice for this creative, experiential learning approach was motivated by our impression during the preparation and exploratory phase in 2014–2015 that many children felt alienated from their bodies, needs, and desires. For example, if we asked the children what they would like to do, nobody would answer the question. They had a hard time expressing themselves, and seemed to find it difficult to connect with their personal desires and needs. Often, they looked around in the class to see what their peers/classmates (authorities) might say, but remained remote and silent. To break through this silence and alienation we offered them sensuous, affective, and emotional activities they had never done before, such as a mindfulness course and a ‘wild pick walk’ (wildpluk wandeling). During this walk through the neighborhood, the children gathered wild edible plants and mixed them in fruit drinks afterwards. Additionally, an important element was the camera given to them, which placed them literally in the position of agent (not being looked at, but pointing the camera to the world; seeing instead of being seen/labelled). The children almost unanimously appreciated activities such as these, and especially liked the fact that they were invited to engage in the activities and inquiries themselves. A few responses about the food lab:
‘Making ice tea was super fun. I added too much sugar (25 cubes). But it tasted good. Later we added colors to it’.‘I like the food lab. We tested our senses by tasting things you normally do not eat. Like warm coke, or cold tomato soup. We did that to test our senses’.‘We did all kinds of tests, like coke with ‘drop’ (coca cola with licorice). Some things we made ourselves. At the end we made a healthy treat. I made a ship with cucumber, cheese, melon and tomato’.(KLIK newspaper, 2019, p. 16)

This heightened their understanding and resulted in embodied knowledge and insights, for example the kids feeling agile while assuming that having the label of being obese would make one stiff and weak, or discovering they liked vegetables after assuming the taste would be gross.

This active, experiential, embodied, and explorative approach in relation to health-related topics was a completely new experience for them; many parents did not have the resources to offer these experiences to their kids, and teachers were often too busy with educational tasks and controlling the classroom.

#### 3.2.2. Working Principle 2: Addressing Uncomfortable Issues

In KLIK, we deliberately did not shun topics the kids thought of as being weird, scary, or embarrassing. We touched on topics falling outside their comfort zone. We noticed for example there was a lot of shame among the children. It seemed as if the children grew up with shame and aversions to their bodies. Many topics were therefore not touched upon at home or remained hidden and were silenced. For example, after visiting a museum about the body (Corpus), we asked the children what they thought of as stupid, dirty, or weird. Many of the girls answered ‘the womb’ and many boys said ‘sperm’. At the same time, they also specifically mentioned these subjects when we asked them if they had learned anything new from visiting the museum. The children, who were approximately 12 years old at that time, seemed to have very limited knowledge in these areas, as the information about fertilization and birth appeared to be new for many of them.
While walking into the giant womb in the museum, one of the girls who had already entered puberty said, astonished: ‘So this is what I have in my body!’(Logbook, 22 January 2019)

Another topic that had remained hidden was being overweight. While many children were labeled as being overweight or obese, this was not an openly discussed.

In our lessons, we addressed those issues, but in a light and positive way. We tried to show the children that the human body is an exceptionally complex and beautiful thing that is worth taking good care of. We sought to approach uncomfortable issues in a humorous way and by using creative methods. For example, when one of the teachers shared with us that many children suffered from stomach pain and constipation we addressed this by making poop out of peanut butter and ginger bread (see Figure 6). We combined this with information on fiber found in vegetables and fruits, and the necessity of drinking water. Later, they proudly shared their insights on a podium and in a newspaper made by kids for kids in the whole school. One of the girls recalls during the evaluation of KLIK:
‘I made ‘poop’ and tasted it. It was weird, and I would never have come up with such an idea. It tasted good because it was made of ginger bread and peanut butter. Making it was quite messy, but it tasted good. It sounds strange but it was yummy. It was weird and funny, and it stuck to your teeth. But it was healthy. I didn’t know about different forms of poop, and thought it was really fun’.(KLIK newspaper, 2019, p. 20)

Mindfulness was also something many children thought of as very strange. Two specialized trainers visited and invited the children to be silent and learn to attend to their breathing and other physical sensations as an anchor for experiencing the here-and-now. The children were trained to turn their attention inward and to focus on their own thoughts and emotions instead of what was going on around them. The children learned they had to practice and train their ‘attention muscle’ to become stronger as a person (see Figure 7).
The mindfulness trainer noticed that over the course of the program moments of silence in the classrooms increased. In lessons six and seven the majority of the kids attend to the instructions without music for more than a couple of minutes. One incident shows the impact: There was a child with a tooth that came loose. He had blood on his hands, but he stayed where he was. At the end of the lesson, the tooth had come out. He went to the bathroom, rinsed his mouth, and came back to the class with a clean tooth. The lesson was not disturbed by this incident, there was no need to involve the other children in the issue, and this particular child was able to bring his attention back to the focus of the moment the attention was brought back to the focus of the moment.(Logbook, 19 March 2018)

Responses of the children on the mindfulness training:
‘If I am quiet, I notice the class is quiet’.‘If there are no thoughts in my mind, it becomes quiet in my head’.‘I cannot think ‘do not think’, because if I think ‘do not think’ I think’.‘I can feel my breath going through my body’.‘It was quite difficult not to respond to the soap bubbles’.‘When I had to go to the dentist I had a strange feeling in my throat. Then I imagined the book and thought of the picture I liked so much. Then that feeling went away’.(Logbook, 19 March 2018)

#### 3.2.3. Working Principle 3: Stepping Outside Your Environment

Many of the children never leave their neighborhood. Stepping outside their environment can be inspiring and motivating. In KLIK, a lot of activities took place outside the school, many activities were planned in B.a.d. (a collective of scholarly artists who live and work in an old school building in the neighborhood), and a few took the kids outside their neighborhood. The following conversation, which took place during a plant-lab at B.a.d. during which the children cut plants to study their growth as a metaphor for their own growth, illustrates why this was meaningful:
A couple of boys are sharing their wonder about the location and have a discussion: One of them says that there are people living in this building, the other cannot belief this and responds ‘No, nobody lives here’ or even ‘No, one cannot live here’. If I say to him that it is true, he is very surprised. Funny to see how a new environment is an enriching experience. This is also true for printing photos themselves. The mini-printers produce the photos in a sequence of yellow, pink, blue, and white. Initially the kids assume the photos failed. We explain the process, and now they understand how colors are built up.(Logbook, 14 February 2019)

This enriches the children’s experiences and is important because their scope is very narrow. Many children do not even leave their neighborhoods. Parents do not have the resources for making ends meet. One of the highlights is the visit to Corpus, a museum on the body, in year three. The logbook fragment below offers an impression of the responses:
I am with one of the teachers in group 1. She acts irritated, and is quite strict and negative; she speaks in an angry voice and corrects the kids all the time. I have to admit that the group starts quite noisy and busy on our trip, but gradually they become more quiet. The teacher becomes more relaxed, and now and then says to the kids ‘Interesting isn’t it?’ Some immediately see that we enter the museum through the asshole. Two girls come to me and say this is not logical; the month should be the entrance. If the information is about sperm the children giggle, but once in the womb all the children are silent and listen to what they are told. When we enter the womb one of the girls says: ‘So this is what I have in my body!’ Most of the time the kids are interested. At the end one of boys whispers in my ear: ‘What do you think of it?’ ‘I like it. Do you?’ Yes, I reply, and what about you? ‘I do’ he replies. On our way back home in the bus I hear the children talk about the enormous private villas in Wassenaar (one of richest areas in The Netherlands). (Logbook, 22 January 2019)

The logbook fragment above illustrates how the children are enriched by entering a new environment. Below are examples that show how children were also stimulated to look at their own, usual environment in a different way. This was the case with the wild pick walk and photography tour. Below are some of the children’s responses to the photography tour and how this changed their perceptions of their environment (see also Figure 8):
‘The photography tour was fun because you stop at places you normally do not notice’.‘I liked the photography tour because I became more aware of my environment’.‘The tour took us through the nature in our neighborhood. I really like being in nature’. ‘Seeing more nature’.‘Usual things become more interesting’. (KLIK newspaper, 2019, p. 18)

#### 3.2.4. Working Principle 4: Keeping It Simple

Being and staying healthy is a very complex matter in which many things are interrelated and difficult to influence. We continuously sought to make this comprehensive subject understandable for children, and to motivate them to act. Yet, the power and control of children, especially in this age group, is limited. In KLIK, we helped them to research and discover what (no matter how small) steps they themselves could take to take care of their own health and well-being. We deliberately started with a simple message: Living healthy = eating healthy and being physically active. This meant breaking complex subjects into small, manageable parts. What we offered was practical, concrete, and visible. Even though we tried to start as simple as possible, we sometimes sensed our message was still too complicated. We then responded by making it even more simple. This required us to be continuously flexible and to tune in into the learning needs of the children. The following example illustrates this (see Figure 9):
Today we work on the Food Guide. The teacher has already downloaded the food circle on the screen and asked the children questions about it. This works well because the children are quite smart and handy in placing products on the poster. Yet when we ask what belongs to what there are a few products that cause problems. What about a rice waffle, and is cheese a fat or carbohydrate, and what about peanut butter? If I ask them if they want to know more, they mention things like baking our own cookies with sugar, butter, jelly, and chocolate. Making it yourself is healthy, or what? And what about chocolate paste with hazelnuts? Those nuts are healthy, aren’t they? A boy asks what happens if you never eat something from the food circle, will you die? It is difficult to frame the message in a way that is not just negative. That one should eat from all segments of the circle, but not too much. A girl asks if you do not like something, what should you do? I propose that you than choose something from that segment that you do like. I give an example: ‘Imagine you do not like milk, then you may choose yogurt or cheese. If you do not like endive, you may choose spinach’.(Logbook, 21 March 2017)

Our goal was to inspire the children with concrete examples of simple, funny, and tasteful healthy lifestyle adjustments. In retrospect, we see how we sometimes made it too complicated. Complex messages did not reach the children, and we therefore had to continuously remain open to their responses and adjust our messages. Repetition also proved to be extremely important. What also worked quite well was to ask the children themselves to explain to each other why we did a certain activity and why it fit into the KLIK program. This resulted in a new pattern over the course of a longer time frame. This also concerned the parents. A good example is drinking water:
In a conversation with the parents, they brought to the fore how they liked the water drinks: Water with cucumber, with lemon, and with mint and raspberry. ‘Amazing, very delicious!’ one of them shared. A few parents recognized it and made it themselves. A father: ‘I will surely make this at home’. A mother: ‘This looks awesome, also for a party!’ Still another mother told us that she would use water, lemon, mint, and sugar, then boil it to make home-made lemonade. How that is relatively easy, cheap, and healthy, because it contains less sugar then sodas.(Logbook, 17 July 2017)

Over the three years, the relationships and contact with the children were strengthened, which was in itself important, but also enabled us to better adjust the activities to their needs and capabilities, and to build upon the knowledge they gained from the program. For example, in year one we started very simple and with activities related to the short-term. In year two, we organized a sport lab in which they experienced how you will sweat if you start exercising or how your heart starts beating more intensely. Later, in year three, based on all the knowledge they had gained over the years and being older we worked on more complex and abstract notions of health and well-being with a long-term effect such as growing muscles and improving one’s condition and agility. The timeframe and layered approach of KLIK enabled us to grow alongside the group of children from Group 6 to 8, making our messages over the years more comprehensive. We saw how this worked well for kids who understood the message easily in year one, as well as for those who did not begin to understand everything until the third year.

## 4. Discussion

This paper aimed to contribute to the understanding of the working principles and impact of PHR with children. We presented KLIK as a health program for school kids that emerged through conversation with the children, teachers, and other community partners. Over the period of four years, the relationships and contact with the children were strengthened, and this enabled us to adjust activities to their needs and capabilities, and to build upon the knowledge they gained through the program. The results demonstrate increased knowledge, awareness, and repertoire of skills in relation to health and well-being among the children. Children, parents, and partners appreciated its creative and experiential approach. KLIK brought about an impact in terms of reaching and engaging many children, parents, teachers and community partners, and in broadening and deepening the children’s awareness and understanding of what it means to live healthy and well. Metaphorically speaking, we saw seeds planted in many children that we believe will help them to grow into healthier adults. Central working principles and meaningful ways to involve children in health-related research include: (1) Experiential learning; (2) addressing uncomfortable issues; (3) stepping outside your environment; and (4) keeping it simple.

Regular health promotion interventions inside and outside schools are often experienced as normative and patronizing, and are based on linear notions of lifestyle changes. Such interventions do not take into account the context of why it is important to change lifestyles and habits and what someone can do, given their circumstances (poverty, illiteracy, cultural habits). This evaluation shows that complex changes do not follow such a pathway, but are multilayered, capricious, highly individual, and contextualized [40,41]. KLIK and its working principles form an answer to the shortcomings of linear programs. It was successful but, and perhaps due to the fact that it was not carried out as planned, many preordained ideas proved to be too ambitious which showed that plans and ideas can be good but too linear because the day-to-day reality in a neighborhood and at school is often much more complex than imagined. It required constant adjustments and flexibility. Through our lessons, we aimed at teaching children to see the bigger picture by gradually zooming out: We started by teaching them the more basic principles of healthy eating and exercise, and through time we focused more and more on why this is important. We sometimes struggled with how to do this properly: It is much easier to say ‘you must do this and you should not do that’, than to explain why. This also meant that we needed to stay alert to our expert-driven solution-reflex as researchers. Being present, giving support, and listening to the responses of the children, teachers and parents appeared much more important and fruitful, and in line with a care ethics approach to education [26,42,43].

Working in a participatory way with the children was challenging. One of the issues was that the children were very used to being told what to (not) do, but were not used to being asked about their opinions on things. This has been reported in the literature on families with a low SES and schools attended by this population [44,45]. Raising awareness and empowerment were therefore crucial in KLIK and it took quite some time to develop a relationship of trust wherein the children felt safe and invited to raise and find their voices. We have seen that the first step to empowerment of children in KLIK was the decision not to ask their parents or teachers, but to approach the children themselves to come up with ideas [46]. Giving them a camera symbolized this step: They were literally in control of what was seen and pictured instead of being objectified by the gaze of adults [47]. Being asked and consulted, and being listened to was a new experience for them, and this interest and appreciation of their ideas was in sharp contrast to their formerly internalized, often negative experiences with adults who did not have a real and genuine interest in who they were and what they felt and thought [48].

While KLIK had impact on the lives of the children, this was not easily measurable in terms of objective outcomes such as a lower BMI or increased fitness. A dilemma we experienced was that the funding agency expected measurable outcomes as a legitimation to continue KLIK. The funding agency had preordained weight loss as an outcome parameter of the programs. The schools expected clear practical guidelines on how to continue KLIK instead of working principles that offer more room to adjust activities for groups of children in the near future. The children also found it sometimes difficult to experiment. Trial and error were seen as tricky because failure was not conceived as a learning experience. They were also part of a culture that focused on measurable success [26]. Over the course of the program, however, the children learned to appreciate this as a ‘special’ and distinctive aspect of KLIK, and the teachers and parents also valued this ‘extra’ they could not offer the kids. This is the strength and weakness of a program that does not readily fit into the school context and culture: It adds important elements because of creative, free, and experiential nature, but is also hard to integrate because it is so different.

The structural implementation of (elements of) KLIK inside the schools and partner organizations prove indeed to be challenging. This is not only related to organizational and financial limitations, but also in relation to the key working principles. To some extent, these working principles conflict with the structural implementation in contemporary organizations. School programs are overloaded, and activities cannot just be adjusted. Community partners also have their own organizations, goals, and conditions. Moreover, the KLIK program cannot just be translated one-to-one in another context. Integration in schools is desirable but also complex, partly because KLIK emerged over time in conservation with all stakeholders (and thus was not a fully preordained and planned program). This was in part because the teachers do not have time to implement the program. We therefore recommend the implementation of the underlying working principles in line with the values underlying these principles.

## 5. Conclusions

Our evaluation of a participatory project such as KLIK shows the value of a creative, flexible approach that is complementary to a regular educational program because it is different. The participatory approach enabled the children to renew their perceptions of self and their environment, to enrich their experiences, and to explore and discover new preferences and talents. The children in the evaluation study appreciated the ‘special’ nature of KLIK and the freedom to inquire and explore what it means to live healthily and well. The working principles include experiential learning, addressing uncomfortable issues, stepping outside your environment, and keeping it simple. The mix of activities is needed to show that living healthily extends beyond health food and exercise; awareness, connection with yourself, and empowerment are crucial to change one’s lifestyle. This way of working is time consuming, more abstract, and less tangible and measurable, but may over the long run be more sustainable. Based on our evaluation of KLIK, we recommend schools and health promotion organizations consider a more holistic approach in which activities related to health and well-being are better integrated and jointly developed with children and community partners in order to be able to tune in to the context, needs, and capabilities of everyone involved.

## Figures and Tables

**Figure 1 ijerph-17-00451-f001:**
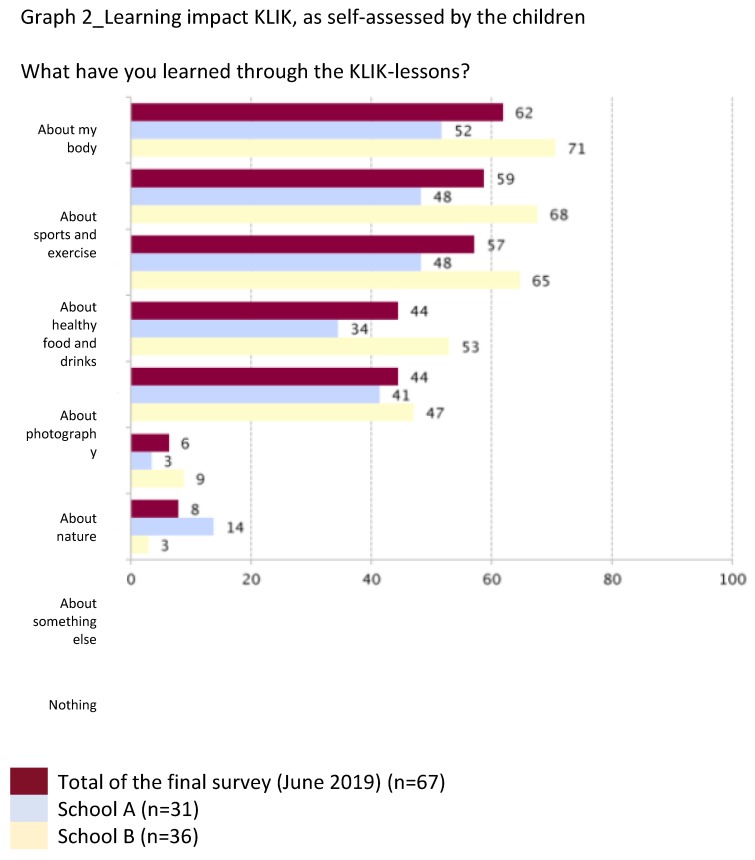
Learning impact of KLIK.

**Figure 2 ijerph-17-00451-f002:**
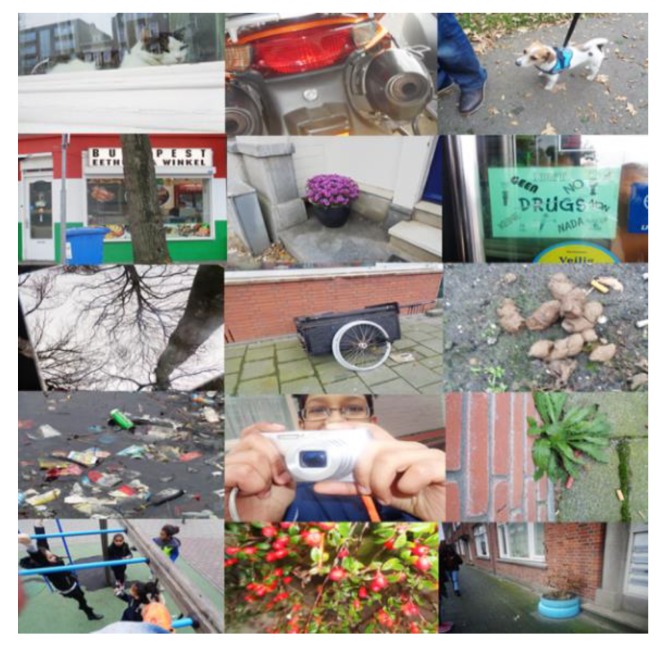
Pictures made by children of their neighborhood.

**Figure 3 ijerph-17-00451-f003:**
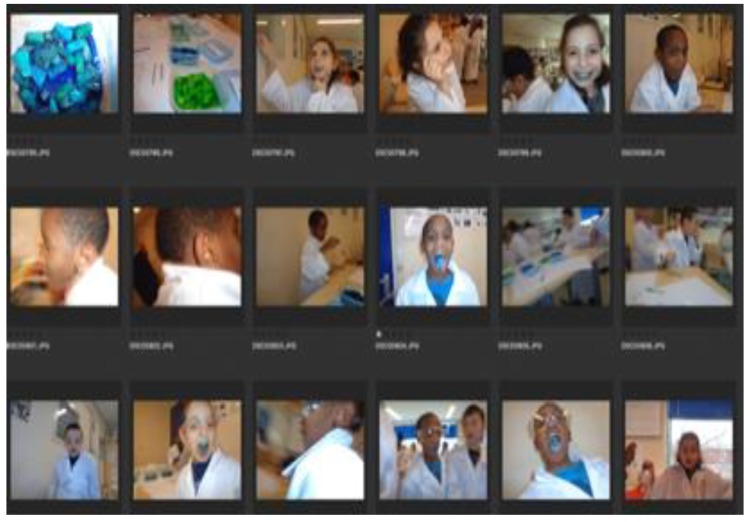
Selfies of blue tongues at the end of the food lab.

**Figure 4 ijerph-17-00451-f004:**
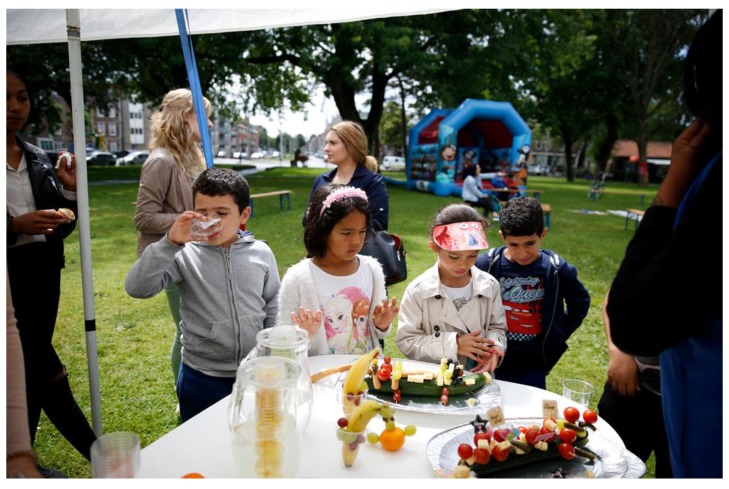
Eagerness to taste healthy treats at the Beach week.

**Figure 5 ijerph-17-00451-f005:**
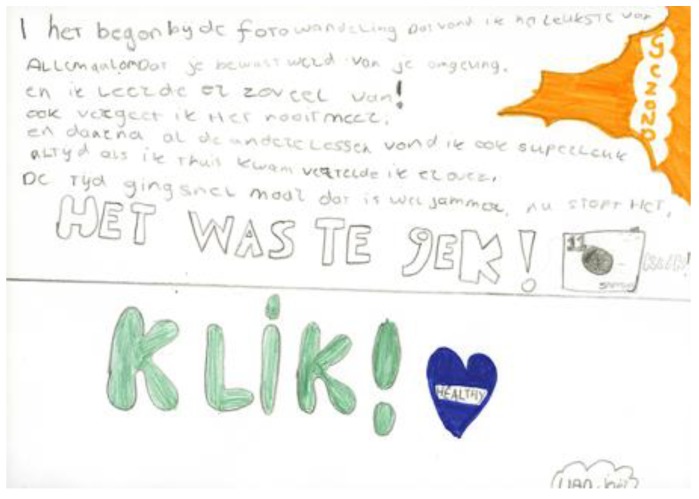
KLIK was te gek! (KLIK was awesome!).

**Figure 6 ijerph-17-00451-f006:**
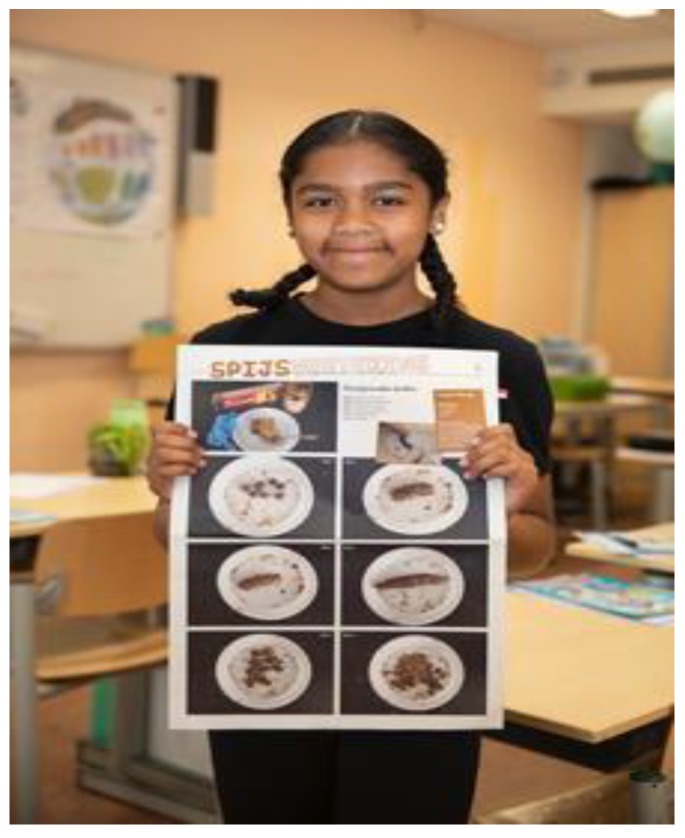
Making poop out of peanut butter and ginger bread.

**Figure 7 ijerph-17-00451-f007:**
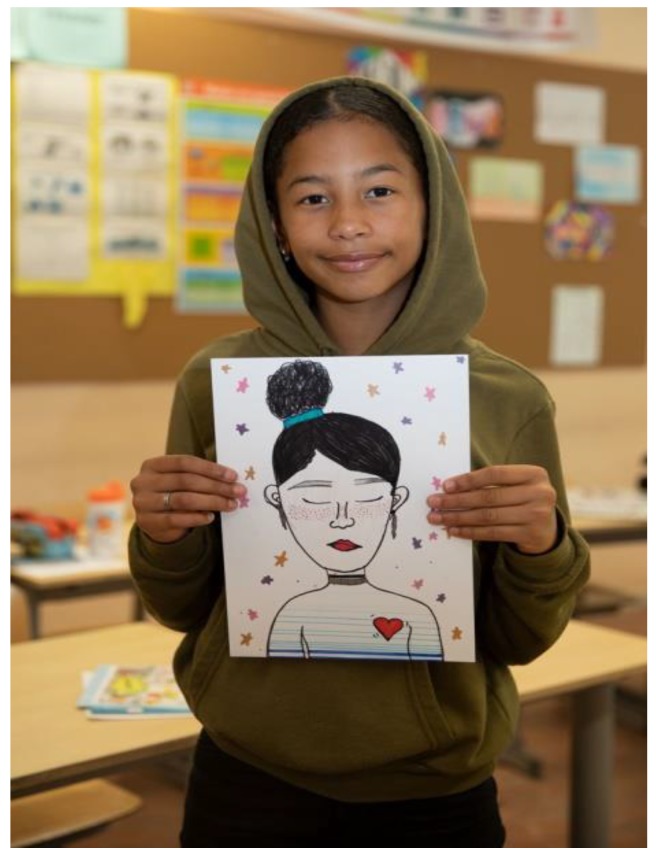
Mindfulness was favorite lesson.

**Figure 8 ijerph-17-00451-f008:**
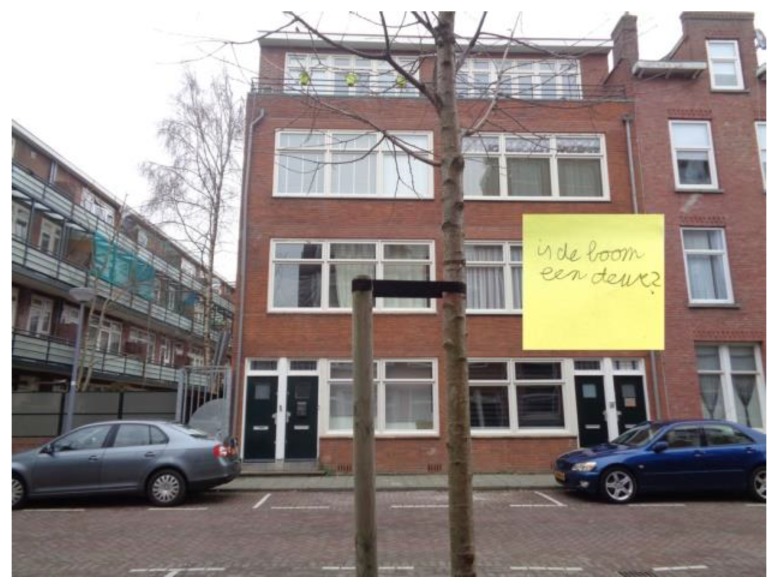
‘Is the tree a door?’.

**Figure 9 ijerph-17-00451-f009:**
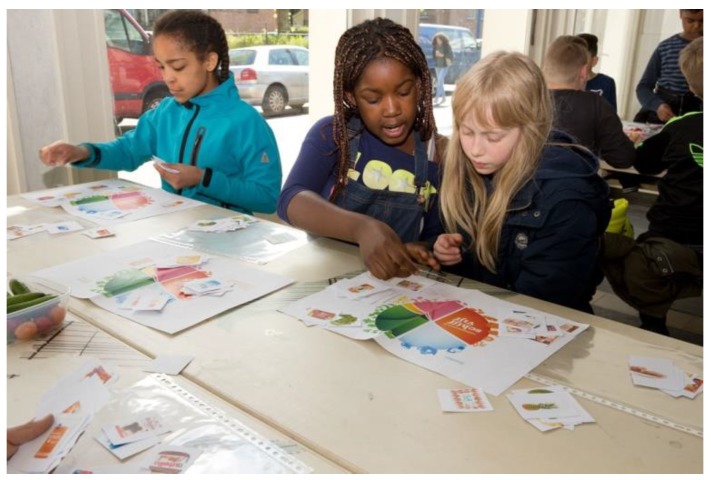
Playing with and placing products on a poster.

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
