# Peer review of "Sowing Seeds to Harvest Healthier Adults: The Working Principles and Impact of Participatory Health Research with Children in a Primary School Context"

_ijerph, 2020, doi:10.3390/ijerph17020451_

Round 1

Reviewer 1 Report

Thank you for your submission of this interesting article. Participation and participatory action research is an area of huge benefit to researchers and often under published, so I welcome the submission and subsequent publication of this work. I like the transparent nature of your article. I have few queries for you to address please. 

Query

1) what were the interview questions/ instructions given (perhaps supplementary doc would be useful to provide context such detail)

2) line 134- developed a short on-line survey for the children and their parents to  measure their perceived fitness and health-  how was this developed/ what questions? did you test reliability for example?

3) e.g. line 173 " because we noticed that the children associated being healthy with fruit and vegetables but did not link it to sport and 
 exercise. We also offered them a training Mindfulness in year two because we observed that many  children were having troubles to keep their attention and concentration in class."

Please can you provide further detail on your methodology in terms of how you have observed, who, and how did you analysis this to make decisions. such as how did you 'notice' 'observe'- do attention and concentration was measured/monitored in what way for example.  

4) as part of a participatory research process, which has involved a range of methodological approaches, please can you provide some information on the 'research and intervention teams' who were they and what input/ background did they bring to the project and evaluation of the project- did you have consistency over the project period or did this change (if so did this have an impact on the behaviour/feedback from the children and stakeholders?).  For example in qualitative methodology we would be transparent in our backgrounds and approach towards analysing the data. (involvement with obesity agenda for example, experience working with children, prior or current relationships with the schools )

4) you have address the ethical approval, but can you flag if during the project period there were any ethical issued that needed addressing.  If yes then be clear how managed, if not then this is also important to rely to the readership that this approach is feasible. 

Minor

para around lines 122.  insert the word old after years please e.g. (8-9 years old)  

Author Response

Dear reviewer,

Thanks for the time given to my manuscript, and the constructive feedback. I have taken all the suggestions on board, and tried to respond to the questions you have raised concerning the methods.

Below you find how I tried to address your comments.

Thanks again for raising important questions. I feel this has helped me to improve the manuscript.

Kind regards,

Tineke Abma, also on behalf of the other authors

_______________________________________________________________

Overview of responses to points raised

Thank you for your submission of this interesting article. Participation and participatory action research is an area of huge benefit to researchers and often under published, so I welcome the submission and subsequent publication of this work. I like the transparent nature of your article. I have few queries for you to address please.

Query

what were the interview questions/ instructions given (perhaps supplementary doc would be useful to provide context such detail)

Response: We have added the topics for the interviews in the methods section (line: 143-146)

line 134- developed a short on-line survey for the children and their parents to measure their perceived fitness and health- how was this developed/ what questions? did you test reliability for example?

Response: We included more information: line 210-217 on the survey and its topics.

3) e.g. line 173 " because we noticed that the children associated being healthy with fruit and vegetables but did not link it to sport and
 exercise. We also offered them a training Mindfulness in year two because we observed that many children were having troubles to keep their attention and concentration in class."

Please can you provide further detail on your methodology in terms of how you have observed, who, and how did you analysis this to make decisions. such as how did you 'notice' 'observe'- do attention and concentration was measured/monitored in what way for example.  

Response: We added information on the participant observations: lines 191-194. And we extended the paragraph on the analysis: line 196-206.

as part of a participatory research process, which has involved a range of methodological approaches, please can you provide some information on the 'research and intervention teams' who were they and what input/ background did they bring to the project and evaluation of the project- did you have consistency over the project period or did this change (if so did this have an impact on the behaviour/feedback from the children and stakeholders?). For example in qualitative methodology we would be transparent in our backgrounds and approach towards analysing the data. (involvement with obesity agenda for example, experience working with children, prior or current relationships with the schools )

Response: We added information on the analysis: 196-206. Also, did we explain the researcher credentials and background of the team and its composition: line 115-125.

you have address the ethical approval, but can you flag if during the project period there were any ethical issued that needed addressing. If yes then be clear how managed, if not then this is also important to rely to the readership that this approach is feasible.

Response: We added information on the ethical issues and how we dealth with them at the end of the methods section: line 226-230.

Minor

para around lines 122. insert the word old after years please e.g. (8-9 years old)  

                Response: the whole text has been edited, this point has been included: line 156.

Reviewer 2 Report

Is it tomato soup or tomato soap? (Line 364). What is coke with drop? (Line 366). Indicate whether the ethical board approved exposing kids to uncomfortable issues mentioned in Lines 377 to 389. Is it ethical to let young children make poo and even taste it? (Lines 396 to 408).

Author Response

Dear reviewer,

Many thanks for the time given to my manuscript and the constructive feedback. I have taken all your suggestions on board. Below is an overview of I addressed your questions.

Thanks again for raising important issues. I feel this has helped me to improve the manuscript.

Kind regards,

Tineke Abma

--------------------------------------------------------------------------------------

Overview of responses to issues raised:

Is it tomato soup or tomato soap? (Line 364). Soup. (Line: 529)

What is coke with drop? (Line 366). Coca cola with licorice (Line: 531).

Indicate whether the ethical board approved exposing kids to uncomfortable issues mentioned in Lines 377 to 389.

Is it ethical to let young children make poo and even taste it? (Lines 396 to 408).

Response: The ethical board approved the whole project. The children could always withdraw from activities which they sometimes did (in the case of Capoeira), but not in the case of making poo of ginger bread and pinut butter. I have added a few sentences at the end of the methods section: 226-230

Reviewer 3 Report

Thank you for letting me take part of this article concerning a very important aspect within health promoting research. The study is very well written and has a high significance and general interest.

However, I have some minor concerns:

“Data were analyzed interpretatively and crosschecked by the researchers.” I would have liked to have more detailed information regarding the qualitative analysis especially since the material is so extensive and multifaceted.

3.1. The emergence of the KLIK program seems to have a better fit placed within the materials and methods section.

Concerning the result section my real interest is evoked when reading 3.3. Working principles. The section labelled 3.2. Impact is quite lengthy and in bits feels a bit “unanalyzed” and might have benefitted from a higher degree of abstraction.

I hope that these comments will be useful to you as you continue your important work concerning participatory health research.

Author Response

Dear reviewer,

Many thanks for the time given to my manuscript, and for the constructive feedback. I have taken your suggestions on board and tried to attend to the questions. 

You raised some important points concerning the methods and results section, and I feel these have helped me to improve the manuscript.

Below you find an overview of the adjustments.

Kind regards,

Tineke Abma, also on behalf of the other authors

-------------------------------------------------------------------------------------

Overview of the adjustments:

Thank you for letting me take part of this article concerning a very important aspect within health promoting research. The study is very well written and has a high significance and general interest.

However, I have some minor concerns:

“Data were analyzed interpretatively and crosschecked by the researchers.” I would have liked to have more detailed information regarding the qualitative analysis especially since the material is so extensive and multifaceted.

Response: We have added extra detailed information on the analysis: line 196-206

3.1. The emergence of the KLIK program seems to have a better fit placed within the materials and methods section.

Response: Thanks for this suggestion. We have replaced this paragraph to the methods section: line: 160-183

Concerning the result section my real interest is evoked when reading 3.3. Working principles. The section labelled 3.2. Impact is quite lengthy and in bits feels a bit “unanalyzed” and might have benefitted from a higher degree of abstraction.

Response: Agree. We have restructured the impact paragraph: 287-491.

I hope that these comments will be useful to you as you continue your important work concerning participatory health research.